# Metabolomics in Corneal Diseases: A Narrative Review from Clinical Aspects

**DOI:** 10.3390/metabo13030380

**Published:** 2023-03-03

**Authors:** Alvin Wei Jun Teo, Jingwen Zhang, Lei Zhou, Yu-Chi Liu

**Affiliations:** 1Singapore Eye Research Institute, Singapore National Eye Centre, Singapore 168751, Singapore; 2GKT School of Medical Education, King’s College London, London SE1 1UL, UK; 3Department of Applied Biology and Chemical Technology, School of Optometry, Research Centre for SHARP Vision (RCSV), The Hong Kong Polytechnic University, Hong Kong, China; 4Centre for Eye and Vision Research (CEVR), 17W Hong Kong Science Park, Hong Kong, China; 5Tissue Engineering and Cell Therapy Group, Singapore Eye Research Institute, Singapore 169856, Singapore; 6Cornea and Refractive Surgery Group, Singapore Eye Research Institute, Singapore 169856, Singapore; 7Ophthalmology and Visual Sciences Academic Clinical Program, Duke-NUS Medical School, Singapore 169857, Singapore

**Keywords:** cornea, metabolomics, corneal diseases

## Abstract

Corneal pathologies may have subtle manifestations in the initial stages, delaying diagnosis and timely treatment. This can lead to irreversible visual loss. Metabolomics is a rapidly developing field that allows the study of metabolites in a system, providing a complementary tool in the early diagnosis and management of corneal diseases. Early identification of biomarkers is key to prevent disease progression. The advancement of nuclear magnetic resonance and mass spectrometry allows the identification of new biomarkers in the analysis of tear, cornea, and aqueous humor. Novel perspectives on disease mechanisms are identified, which provide vital information for potential targeted therapies in the future. Current treatments are analyzed at a molecular level to offer further information regarding their efficacy. In this article, we provide a comprehensive review of the metabolomic studies undertaken in the cornea and various pathologies such as dry eye disease, Sjogren’s syndrome, keratoconus, post-refractive surgery, contact lens wearers, and diabetic corneas. Lastly, we discuss the exciting future that metabolomics plays in cornea research.

## 1. Introduction

Metabolites are the products and substrates in metabolism that drive cellular functions, either produced by the host organism or derived from microorganisms, diet, medications, or other exogenous and environmental sources. The metabolome is the collection of all low molecular weight metabolites in a biological system and reflects cellular activity and physiological status [1,2]. Recent development has shown that the metabolome can interact with and modulate the genome, proteome, and transcriptome [3]. Thus, metabolomics has become an emerging field in clinical research to unravel the disease processes on a molecular level, as well as develop diagnosis methods and treatment options. In ophthalmology, metabolomics has been employed to study the tissue metabolism and pathogenic pathways in several eye diseases, such as diabetic retinopathy [4], age-related macular degeneration [5], glaucoma [6], and uveitis [7].

The analysis of metabolites from tears or aqueous humor has been used to reveal biomarkers of several ocular diseases such as predicting glaucoma progression [8], with greater sensitivity and specificity than the analysis of plasma or serum [9]. Several corneal diseases or conditions, including dry eye disease [10], keratoconus [11], Sjogren’s syndrome [12], refractive surgery [13], prolonged contact lens wear [14], and diabetic corneas [15], have been reported to have significant metabolomic alterations in tears or corneas. Metabolomics has therefore been applied in the studies of corneal diseases to potentially identify new biomarkers or elucidate the disease mechanisms (Figure 1).

Two general approaches exist in metabolomic profiling: targeted and non-targeted. Targeted approaches identify and quantify selected known metabolites via isotope-labeled internal standards, while the non-targeted approach aims to profile as many metabolites as possible, and the identities of which are not established prior to analysis [16]. Non-targeted profiling allows the discovery of novel pathways, but the complexity of biofluids with a wide range of compound classes and metabolite abundance have intrinsic limitations with current techniques [17]. Nevertheless, advancement in technology and research has led to the building of comprehensive databases such as the human metabolome database [18] to facilitate metabolite identification. 

## 2. Principles of Metabolomic Experiments

There are two principal analytical modalities: nuclear magnetic resonance (NMR) spectroscopy and mass spectrometry (MS) [16]. NMR spectroscopy is a rapid, highly reproducible and non-destructive analytical method that is used to determine the structure of organic compounds. It relies on the ability of spin active nuclei to absorb and re-emit pulsed electromagnetic radiation when placed within a magnetic field [19]. The frequency pattern generated is the consequence of an interaction of the nuclei with the electromagnetic field; this provides information regarding the molecular structure. Hydrogen, carbon, and phosphorus are common targets due to their abundance in biological samples. The advantages of NMR include low cost per sample, minimal sample preparation, allows for analysis of intact tissue, and offers prompt throughput [19]. The disadvantages are that it can be less sensitive than mass spectrometry, requires a skilled user, and has high start-up costs due to instrument acquisition [20].

On the other hand, MS is a destructive analytical process that comprises several key components: the ionization source, mass analyzer, and detector [21]. Molecules first undergo ionization via various techniques such as electrospray ionization or matrix-assisted laser desorption ionization (MALDI) [22]. These ions are then passed through a mass analyzer which separates these molecules based on their mass/charge ratio. Following peak detection and alignment with statistical analysis of full scan data, the metabolites are analyzed and confirmed by comparison of their retention time and spectra with available mass spectral databases [23]. Different separation techniques such as liquid chromatography–mass spectrometry, gas chromatography–mass spectrometry, and ultra-performance liquid chromatography are used based on compound volatility and are complementary to each other. MS is therefore a highly sensitive method of sample analysis. However, its accuracy is highly dependent upon the experimental conditions, sample stability, and the instruments used [24]. Figure 2 summarizes the methodology in corneal metabolomics studies.

## 3. Metabolism and Metabolomic Profiles in Corneas

The cornea requires oxygen to prevent corneal edema and discomfort. In an open eye, oxygen is primarily supplied by the surrounding air. In the closed eye, diffusion from the blood circulation in the palpebral conjunctiva and anterior chamber supplies about two-thirds of the oxygen demand [25]. The two principal metabolic reactions of the cornea are aerobic glycolysis and glutaminolysis via the tricarboxylic acid/Kreb’s cycle [26] and anaerobic glycolysis through the Embden–Meyerhof pathway [27]. Minor metabolic reactions such as the hexose-monophosphate shunt involving the generation of five-carbon sugars, as well as a sorbitol pathway producing fructose and sorbitol, play a minor part in overall corneal metabolism [27,28]. The primary waste products are carbon dioxide and water in aerobic glycolysis, and lactate with hydrogen ions in anaerobic glycolysis [29]. Excess lactate is produced during epithelial hypoxia, especially in the closed eye and during contact lens wear. 

In physiological conditions, water is osmotically driven across the endothelium into the anterior chamber with the Na^+^/K^+^ active ion pump responsible for the outflow of water. When hypoxia occurs, swelling of the cornea occurs due to the imbibition of water from the anterior chamber across the endothelium [30]. As oxygen concentration falls, anaerobic production of lactate ions and buffering by bicarbonate ions increase [30]. Basolateral endothelium osmolarity increases, and the osmotic-driven water efflux rate into the anterior chamber diminishes, resulting in corneal edema as the swelling of the stroma lowers the water leak-in rate that matches the lower pump out rate [30]. 

The metabolic profile of donor corneas in culture media was analyzed in a study comparing warm cultured corneas and corneas that were immediately frozen after excision. The cultured corneas were separated into 3 groups according to their culture periods: less than 9 days, 9–14 days, and 15–20 days [31]. Glucose levels in cultured corneas increased significantly during culturing, possibly due to the degradation of glycogen in corneal epithelium and endothelium. This may prevent cell apoptosis and maintenance of the mitochondrial transmembrane potential in the cornea. The levels of alanine, formate, and lactate were highest in the corneas kept in the culture medium for 9–14 days. The concentrations of acetate, arginine (Arg), choline, glucose, isoleucine, leucine, phenylalanine, tyrosine, and valine were higher in cultured tissues than in frozen corneas, while the reverse was true for ascorbate, glycerophosphocholine, and taurine concentrations. The increase in choline levels could be due to the degradation of cell membranes or the impairment of enzymatic function that controls acetylcholine metabolism. Ascorbate and taurine, which serve as anti-oxidants, were decreased in the cultured corneas in the first week. This is an effect of an increased stress response and taurine’s role in promoting human corneal epithelial cell survival by enhancing membrane stability. Overall, the paper was one of the first to utilize a metabolomic approach in determining the changes occurring in corneas during culture. More metabolomic studies can be performed to determine the optimal cornea storage conditions, such as medium and temperature to ensure long-term survival and stability of corneal grafts.

## 4. Metabolomic Studies in Corneal Diseases

### Dry Eye Disease

Dry eye disease (DED) is defined as a multifactorial disease involving tear film instability, hyperosmolarity, ocular surface damage, and neurosensory abnormalities [32]. Developed mass spectrometry techniques allow us to investigate the tear film at a molecular level to develop novel diagnostic and treatment modalities.

Matrix-assisted laser desorption ionization mass spectrometry imaging (MALDI-MSI) combines high-throughput mass spectrometry and molecular imaging. Tissue sections are covered with a matrix for extracting molecules from the specimen, which aids desorption/ionization for further analysis by the mass spectrometer. Matrix selection and solvents used depend on the analyte class required in imaging. It allows label-free measurement of metabolites directly in tissue sections to provide spatial signatures and insights regarding complex metabolic changes in response to tissue damage [10]. A study utilized this method to analyze different ocular regions in normal controls as well as scopolamine hydrobromide-induced DED in rats [10]. The authors found that glycerophospholipid, essential for cell membrane dynamics and signal transduction, as well as phenylalanine metabolism, were altered in the central cornea and aqueous humor of eyes with DED. The levels of phosphatidylcholine and phosphotidylethanolamine were increased. These are fundamental to tear film stability, and the proper spreading of non-polar tear fluid [33,34]. The increase is attributed to a hyperosmolar environment [35]. Betaine, capable of stabilizing cell volume and inhibiting apoptosis in the human corneal epithelial cell under hyperosmotic stress [36], was over-expressed in the cornea and forniceal conjunctiva. This increase possibly indicates an early cytoprotective mechanism against dry eye. Other studies have shown that administration of betaine in DED ameliorated corneal damage, reduced levels of inflammatory factors [37], and it is currently being investigated as an osmoprotectant to improve cellular homeostasis of the ocular surface [38].

A decrease in Arg in the aqueous humor of DED patients was also reported [10], which is in keeping with previous reports of decreased Arg in tears of DED patients [39]. As Arg contains anti-inflammatory properties [40], a decrease in Arg may lead to increased inflammation in ocular surface disease. Levels of sphingomyelin, ceramide, and glucosylceramide were increased in corneas and aqueous humor with DED. These metabolites have been shown to contribute to inflammation, senescence, inhibition of cell proliferation, and apoptosis, leading to increased ocular surface damage in DED [41]. The study, therefore, demonstrates the potential of MALDI-MSI in providing analysis of multiple metabolites in different ocular tissues and may facilitate the development of individualized clinical interventions in the future.

Managing dry eye disease with serum eye drops has been attempted as serum eye drops are physiologically similar to human tears [42]. Autologous peripheral blood-derived serum (PBS) and cord blood serum (CBS) were compared using NMR spectroscopy to identify the differences from human tears and the molecules for ocular surface healing [43]. Metabolites in both CBS and PBS are much more concentrated than those in normal human tears, except for pyruvate, taurine, and urea. This suggests that serum derivatives can be diluted several times and maintain a tear-like composition. The levels of several amino acids (glutamine, phenylalanine, and alanine), amino acid precursor betaine, myo-inositol, choline, glutamine, creatine, choline, β-hydroxybutyrate, and catabolites (ornithine and urea), were significantly different between CBS and PBS. Choline concentration was over three times higher in CBS than in PBS. This finding is promising for DED therapy as choline has been shown to improve symptoms of DED [44]. Creatine levels were also doubled in CBS compared with PBS, while its breakdown product, creatinine, was lower in CBS. Creatine plays a role in maintaining adenosine triphosphate (ATP) and acid-base balance, providing anti-oxidant functions and stabilizing plasma membranes [45]. CBS also contained higher levels of citrate, glycerol, and ketone bodies (acetoacetate and β-hydroxybutyrate) with respect to PBS. β-hydroxybutyrate might have a beneficial role in DED as it showed suppressive effects on apoptosis and improved corneal epithelial erosion in a dry eye rat model [46], with a further clinical study demonstrating that β-hydroxybutyrate eye drops may be effective for the treatment of tear-deficiency DED [47]. Another metabolite significantly enriched in CBS was myo-inositol. Inositol signaling is a crucial step in regulating tear secretion [48], with inositol 1,4,5-triphosphate binding with its receptors being an important step in fluid secretion [49]. A lack of inositol 1,4,5-triphosphate receptors (IP3R) showed a dry eye phenotype in a mouse model [50] and is downregulated in patients with Sjogren’s syndrome [49]. Supplementation of inositol may counteract the decrease in IP3R in DED patients with Sjogren’s syndrome. Taken together, serum eye drops contain important metabolites to influence cellular signaling in hopes of alleviating DED.

Another study assessed rats with scopolamine-induced DED to characterize metabolic changes in eyes, urine, and serum for the exploration of biomarkers [51]. The authors found that 2-hydroxyisobutyrate, citrate, and succinate were significantly elevated in the plasma. Citrate is involved in inducing TNF-α and interferon-γ (IFN-γ), which are associated with inflammatory signals [52]. In urine, 12 metabolites such as phenylalanine, phenylacetate, pantothenate, glycine, succinate, methanol, valine, propylene glycol, histidine, threonine, lactate, and acetate, were significantly increased. Among them, glycine and histidine, as well as phenylalanine, succinate, and cis-aconitate, in particular, belong to phenylalanine metabolism and are reported to be associated with inflammation [53]. These results suggest the potential use of serum and urine metabolites as biomarkers of DED.

Fatty acids such as essential polyunsaturated fatty acids (EPUFAs), eicosapentaenoic acid (EHA), and docosahexaenoic acid (DHA) have been considered a promising supplementary treatment for DED, evidenced by the inhibitory effects on inflammatory cytokines and T-cell responses in DED [54]. The tear metabolomic profile of patients with DED had significant differences after nutraceutical supplementation with anti-oxidants, DHA, EHA, and docosapentaenoic acid [55]. Before supplementation, the DED and control groups showed differing basal tear metabolomic profiles, with a difference seen in approximately 50 metabolites such as cholesterol, *N*-acetylglucosamine, glutamate, amino-n-butyrate, choline, glucose, and formate. After supplementation, the metabolomic differences between the DED group and the control group decreased. Choline and acetylcholine in tears had increased after EPUFA supplementation [55]. This may further stabilize the tear film, exert antinociceptive effects on the central nervous system, and suppress ocular surface inflammation [56]. The study demonstrated that the tear metabolomic profile of patients with DED can be modified with appropriate oral supplementation containing anti-oxidants and essential fatty acids.

## 5. Sjogren’s Syndrome

Sjogren’s syndrome is a chronic, progressive multisystem autoimmune condition that is associated with DED. Currently, there is no single test that can distinguish it from other causes of DED. A study sought to derive targeted metabolomic signatures from tears extracted from patients with primary Sjogren’s syndrome (PSS) and DED controls. It managed to establish machine learning architecture that could differentiate newly diagnosed PSS from patients suffering from DED using nine metabolites [57]. Six phospholipids had increased concentration in SS tears due to phospholipase A2 activity, which is linked to dry eye disease through prostaglandin E2 and pro-inflammatory cytokines [58]. Dopamine levels, part of the analyzed metabolomic signature, were decreased due to reduced lacrimal secretion. This depletion has been hypothesized to stem from inflammatory autoantibodies, and this decrease also reduces the blink rate and exacerbates the severity of PSS [12]. Serine, a substrate involved in the process of synthesis of glutathione, an anti-oxidant, was found in a reduced level and might also contribute to the disease severity.

## 6. Contact Lens Wearing

Despite advancements in contact lens technology to optimize lens oxygen permeability, wearing soft lenses impedes oxygen supply to the cornea due to reduced exposure of the corneal epithelium to environmental air. This results in cornea swelling, loss of transparency, limbal hyperemia, and neovascularization [59]. Chhabra et al. have shown that there is a decrease in aerobic metabolism and an increase in anaerobic metabolism during prolonged contact lens wear [59].

Lipids form a crucial area of the tear film, maintaining tear film stability and ocular surface homeostasis. Major components of the tear film include long, saturated chains of wax esters, cholesterol esters, nonpolar components such as triacylglycerols, diesters, free sterols, free fatty acids, and a polar lipid interface between nonpolar lipids and the aqueous layer [60]. Silicon hydrogel lenses result in lipid deposition on the lenses because of their hydrophobic properties and included organosilicon moieties of lotrafilcon A and balafilcon A for better oxygen permeability [61]. This lipid deposition in silicone hydrogel lenses is dependent on the type of material and their equilibrium water content [62], and serves as good collection tools to analyze lipid deposition and understand the metabolomic profile in contact lens users.

A lipidomic study collected tear samples and contact lens extract solvent from patients who wore contact lens daily or continuous wear overnight to determine the tear lipid composition changes during wear. In daily wear lenses, the ratio of unsaturated to saturated lipids was lower than that of continuous wear lenses (1:2 vs. 1:9) with the difference thought to be due to the degradation of unsaturated lipids during overnight wear [63]. This deposition of lipids can result in reduced lens wettability and contact lens discomfort [64]. Another study showed that contact lens wear increased the amount of free cholesterol and phospholipids in tears, but had no effect on cholesteryl, wax esters, triacylglycerols, or (O-acyl)-ω-hydroxy fatty acids [14]. The differences with respect to free cholesterol and the phospholipids are interesting as these lipids are probably not produced by the meibomian glands in the lids [65]. Instead, cholesterol and phospholipids are commonly associated with cell membranes, and it may imply that the change in the concentration of these lipids in tears is associated with damage to cell membranes during lens wear.

Li et al. utilized liquid chromatography to analyze corneal stromal lenticules extracted from small incision lenticule extraction (SMILE) from soft contact lens-wearing patients. They then correlated the findings with the corneal nerve status obtained by in vivo confocal microscopy [66]. Patients were divided into 4 groups based on their duration of contact lens wear: non-wearing, less than 5 years, 5–10 years, and more than 10 years. The central corneal sub-basal nerve fiber density in the non-wearing group was significantly higher than the three other groups. This might be related to the downregulation of neuroprotection metabolites, such as taurine, docosahexaenoic acid, and sphinganine in contact lens wearers, especially in the group which wore lenses longer than 5 years. The constituents of the lipid layer were also decreased, suggesting an impairment of lipid synthesis during contact lens wear, resulting in eye discomfort. Upregulation of inflammatory factors such as arachidonic acid and linoleic acid, as well as short-chain organic acids such as citrate, pyruvate, succinate, oxaloacetate, and glutamate, suggest increased inflammation and dysregulation of the citric acid cycle. These results may explain the eye discomfort felt after wearing contact lenses.

## 7. Refractive Surgery

Refractive surgery has good visual and refractive outcomes, but it may result in postoperative dry eye and even neurotrophic epitheliopathy due to the transection of corneal nerves that provide trophic support to the ocular surface [67,68]. A lipidomic study has shown modifications of the tear film lipid layer after refractive surgery, and this is likely to have effects on its physiological function [69]. A study correlated metabolomic profiles of lenticules derived from corneas undergoing SMILE with the rate of postoperative corneal nerve recovery [13]. The study reported that corneal nerve fiber length and corneal nerve fiber density recovered to the preoperative level within 1 year in the 18–30 age group, but those in the 31–40 and 41–50 age groups remained less than that before surgery. Corneal nerve branch density of the latter group at 1 year also remained less relative to before surgery. Moreover, metabolites that are associated with inflammation were found in greater levels in older age groups. Uric acid, a final breakdown product of purine metabolism, was positively correlated with age. Ascorbic acid, taurine, spermidine, acetylcarnitine, carnitine, tryptophan, and hydroxyproline, which are important anti-oxidants and neuroprotective agents [70,71], significantly decreased with age, resulting in slower recovery of corneal nerves. Arg was found in higher levels in the 31–40 age group compared with the 18–30 age group. Although Arg has been found to decrease oxidative stress as covered previously, the authors theorized in this study that excessive levels might be toxic and induce inflammatory effects [72,73]. Histidine and arachidonic acid, associated with inflammation, were significantly increased in the 41–50 age group compared with the 18–30 age group. Taken together, these findings suggest that there is a higher concentration of inflammatory metabolites and lower concentration of anti-oxidants in older patients, influencing the rates of corneal wound healing and corneal nerve recovery.

## 8. Keratoconus

The pathogenesis of keratoconus results from the interaction of genetic and environmental factors [74]. More recently, oxidative stress is hypothesized to contribute to keratoconus pathology. A study investigating metabolic differences between human corneal keratocytes (HCKs), fibroblasts (HCFs), and keratoconus cells (HKCs), under conventional two-dimensional (2D) cultures and a novel three-dimensional (3D) in vitro model found that in keratoconus cells, elevated levels of lactate with upregulation of lactate/pyruvate ratios were observed, along with a decrease in reduced and oxidized glutathione (GSH) ratios when compared with healthy keratocytes [75]. These indicate that keratoconic corneas undergo significant oxidative stress. A subsequent study by the same group investigated tear metabolites in keratoconus and confirmed the above findings [76]. Additionally, this study presented that the metabolic differences between healthy and keratoconus subjects are independent of rigid gas-permeable lens wear. 

Further studies of corneal buttons from healthy and keratoconic corneas revealed more evidence of oxidative stress in keratoconus. Saturated and unsaturated fatty acids, which play important roles in the processes that regulate inflammation and tissue repair, were downregulated in keratoconus [77]. Gluconic acid, an intermediate in the pentose phosphate pathway and an indicator of oxidative stress, was detected only in keratoconus corneas and not in normal corneas. Succinic acid, involved in the tricarboxylic acid cycle, was downregulated, suggesting a dysregulation with changes in energy requirements [77]. There were no differences in the concentrations of the majority of amino acids between keratoconus and normal corneas [78]. Levels of citrate and acetate were found in high concentrations, along with a lowered ratio of reduced GSH and oxidized GSH, providing further evidence of oxidative stress [79] in the pathogenesis of keratoconus. The authors also reported no major metabolic differences between older, normal corneas (age 61–75) and keratoconic corneas (age 19–27), with a suggestion that there might be accelerated aging of the cornea in keratoconus [78].

Defects in extracellular matrix (ECM) deposition by keratoconus cells derived from the corneal stroma is a defining characteristic of keratoconus pathology [80]. Downregulation of cytosolic Arg in keratoconus cells was thought to result in lesser ECM deposition compared with normal corneal keratocytes in a 3D model [75]. Besides its anti-oxidant effects, Arg plays multiple roles in cell proliferation, DNA replication, and collagen deposition [40] and also serves as a precursor to proline and hydroxyproline, which are both highly present in collagen monomers [81]. Proline and hydroxyproline are major constituents of collagen types I and V, which constitute most of the stromal ECM in the well-organized lamellar structure of the cornea. This is essential for its biomechanical properties and for providing good vision [82]. Arg and hydroxyproline have been found in lower levels in keratoconus, and efforts were made to investigate the effects of Arg supplementation on ECM secretion and deposition by keratoconus cells [83]. It was observed that Arg supplementation led to an increase in Arg levels in healthy controls with only a modest change in human keratoconus cells. There was no increase in proline and hydroxyproline levels, but an increase in collagen type I secretion with no significant changes in type V was found. These results suggest that Arg-mediated ECM secretion may not involve the conversion of Arg to proline and hydroxyproline.

Cornea collagen crosslinking (CXL) effectively arrests the progression of keratoconus [84]. It utilizes ultraviolet A light to excite riboflavin into a triplet state, producing reactive oxygen species. It also induces the formation of new covalent bonds between amino acids to improve corneal biomechanical resistance and stability [84]. In an in vitro study, primary human corneal fibroblasts from healthy and keratoconus corneas were analyzed before and after CXL [85]. Levels of GSH, ascorbic acid, tyrosine, and cysteine, all strong anti-oxidants, were increased after CXL in both groups. Ascorbic acid levels were elevated in keratoconus corneas in relation to healthy corneas. There was a significant decrease in lactate levels, and an increase in ATP in keratoconus cells after CXL compared with those before CXL, indicating improved cellular metabolism after treatment. There was also an increase in collagen intermediates after CXL in the keratoconus group, potentially allowing for proper collagen synthesis and assembly. Pro-inflammatory metabolites, such as myo-inositol and histidine, were significantly downregulated in keratoconus corneas after CXL. Overall, CXL in the keratoconic cornea improved the corneal microenvironment with increased anti-oxidant levels, and reduction in lactate and pro-inflammatory metabolites [85].

The tear metabolic status of patients who had progressive keratoconus was also examined before and after CXL. An increase was found in 16 organic acids and a decrease in 8 organic acids 6 months after CXL treatment [86]. The organic acid that showed the greatest percentage increase before and after CXL was N-acetyl-L-aspartic acid, a derivative of aspartic acid that has anti-oxidant effects. The level of 3-OH butyric acid, a marker of impaired ketone body and glucose metabolism, was significantly decreased, suggesting that corneal glucose metabolism improved after CXL. There was a decrease in the lactate/malic acid ratio, indicating normal functioning of the citric acid cycle after CXL as malic acid is an important mediating metabolite in the citric acid cycle.

## 9. Diabetic Corneas

Diabetes is a disease of dysfunctional metabolism, and hyperglycemia leads to excessive production of advanced glycation end products, polyol flux, and reactive oxygen species [67,87]. These substrates further affect corneal epithelium, keratocytes, and nerves, resulting in diabetic keratopathy [88].

Patients with type 2 diabetes mellitus (T2DM) revealed significantly higher levels of carnitine, nicotinic acid, and sorbitol in tears, with carnitine being an endogenous metabolite that was increased in serum [89]. Amino acids such as aspartic acid, glutamate, glutamine, methionine, methionine sulfoxide, serine, threonine, tyrosine, and valine were in higher levels in the tears of T2DM patients, in line with previous diabetic metabolomic studies [90]. These amino acids have been postulated to act via the same pathways as insulin, namely, the activation of the mammalian target of rapamycin (mTOR) and its downstream targets [91]. The accumulation of amino acids may also cause mitochondrial dysfunction via stress kinase stimulation, resulting in β-cell apoptosis [92]. Both effects may lead to insulin resistance and T2DM [92]. Raised levels of uric acid in tears was also found, similar to high uric acid levels in serum. This is common in diabetic patients and appears to contribute to insulin resistance by the development of mitochondrial oxidative stress and impairment of insulin-dependent stimulation of nitric oxide in endothelial cells [93]. Hence, the raised levels of uric acid in tears could either be due to diabetic nephropathy or contribute to diabetic change in the eye. Taurine has several important functions, including membrane stabilization, anti-oxidation, osmoregulation, and pro-inflammatory properties. The accumulation of taurine in tears of T2DM patients has been similarly described in cases of dry eye syndrome [94], and this increase could reflect an imbalance of metabolic homeostasis due to long-lasting stress and inflammation of the ocular surface [95], with authors suggesting that it could be a novel tear biomarker of diabetic patients. On the contrary, serum levels of taurine in diabetics have been found to be reduced, and taurine deficiency was found to be linked to diabetic complications such as diabetic neuropathy [96].

A study analyzing healthy corneas, type 1 diabetes mellitus (DM), and T2DM diabetic corneas from cadavers attempted to determine if glucose, lipid, and amino acid metabolism were altered in the diabetic corneal stroma [97]. Glyoxylate, touted as a biomarker for early detection of T2DM [15], as well as citric acid cycle intermediates, were little changed in diabetic corneas compared with controls. This suggests that glucose derivatives may not be useful biomarkers in diabetic corneas. Lipid metabolism was altered with significant upregulation of sphingosine and dihdrosphingosine, which play important roles in the insulin signaling pathway [98]. In protein metabolism, most amino acids and derivatives were little changed among the groups. However, there was an increase in the tryptophan derivative, indole-3-carboxylic acid, in T1DM corneas. Tryptophan is an essential amino acid that is metabolized mainly to kynurenine and nicotinic acid to generate nicotinamide adenine dinucleotide (NAD), an essential substrate in the glycolysis and citric acid cycle. In T1DM, there was an upregulation of xanthurenic acid and kynurenic acid in the corneal stroma, suggesting dysregulation of kynurenine metabolism. Kynurenic acid is known to have antioxidant, anti-inflammatory, and anti-proliferative properties [99,100] and the kynurenine pathway has been linked to corneal endothelium health [101]. In T2DM, however, there was no increase in kynurenic acid and xanthurenic acid but a significant increase in nicotinate levels due to the modulation of tryptophan metabolism, with no increase in the metabolites in the kynurenine pathway. Other significant findings include an increase in type III collagen in T1DM and T2DM corneas [102]. This might be a result of prolonged exposure to a hyperglycemic environment, and a subsequent abundance of reactive oxygen species, leading to keratocyte activation and subsequent fibrotic activity.

The effect of corneal nerves on the metabolic function in the diabetic corneal stroma was also investigated. In a study involving 3D in vitro constructs with and without harvested fibroblasts and corneal nerves [103], samples were separated into healthy, T1DM, and T2DM groups, respectively. When comparing T1DM and T2DM innervated groups versus T1DM and T2DM non-innervated groups, it was found that the introduction of neurons influenced the number of metabolites in healthy controls and the respective diabetic groups. The pathways that were affected by the addition of neurons for both T1DM and T2DM groups were pyrimidine metabolism, purine metabolism, aspartate metabolism, and methionine metabolism. When comparing healthy controls with and without innervation, the authors found that the addition of neurons supported pathways such as pyrimidine metabolism, glycerol phosphate shuttle, electron transport chain, and glycolysis. These are amino acid and nucleic acid metabolic pathways that are involved in cellular metabolism and signal transduction, and our results, therefore, highlight the essential role that neurons play in these metabolic pathways. When comparing innervated T1DM cultures with healthy innervated cultures, there were alterations in key energy production metabolic processes, such as phosphatidylethanolamine biosynthesis, pyrimidine metabolism, methionine metabolism, and aspartate metabolism. For innervated T2DM and healthy innervated cultures, there was a difference between aspartate metabolism, glycerol phosphate shuttle, electron transport chain, and the gluconeogenesis pathway. Interestingly, there was an upregulation in the majority of biosynthetic amino acids and phospholipid pathways in T1DM compared with T2DM, with 83% of metabolites at lower levels in T2DM. Arg and glutathione were reduced in T2DM cultures, suggesting increased levels of oxidative stress. These suggest a possible difference between specific biochemical processes of T1DM and T2DM. Despite the above-mentioned differences, some metabolites, mainly in the glycolysis pathway, were found to present in similar patterns, indicating a similar dysfunction in their glucose metabolism. Myoinositol, thought to be involved in the promotion of insulin signaling, was upregulated in both T1DM and T2DM constructs, and it might be related to neuronal dysfunction in diabetic corneas. 

Table 1 summarizes the studies reviewed in the literature, their findings, and clinical implications.

## 10. Conclusions and Future Direction

In this review, we summarized the literature on the metabolomics of corneal diseases and demonstrated that the metabolomic profile of corneal tissue and tears are valuable sources in the study of corneal diseases. Metabolomic signatures have the potential for personalized medicine, with advances in technology allowing us to delve deeper into the understanding of individual metabolic profiles in corneal diseases. The role corneal nerves play in the cornea metabolic environment sheds another light on the importance of nerve morphology. However, several challenges remain. Although investigations of cornea metabolomics have yielded valuable information regarding oxidation processes, disease influence on metabolism and inflammatory metabolites, and the varying severity of disease in the study population, could influence study results. There is a need to standardize study techniques and patient populations to achieve consistent results. This would allow clinicians to acquire a better understanding and lead to the development of methods to improve the diagnosis and management of the disease. Being a nascent field, many questions remain regarding the metabolite profile found in these corneal diseases. However, novel biomarkers derived from these studies may be a promising initial start in the diagnosis and management of these diseases, with the ability to measure changes on a molecular level during the disease progression. With larger sample sizes and longer follow-up periods to attest to the validity, these may present an exciting opportunity for early diagnosis and alternative management of corneal diseases. In addition, it is hoped that with more research, it will be possible to construct a library of metabolomic signatures of the cornea to facilitate the understanding of pathogenesis and treatment efficacy. Metabolomics has a promising future and will continue to play an important role in the clinical applications of these corneal diseases. 

## Figures and Tables

**Figure 1 metabolites-13-00380-f001:**
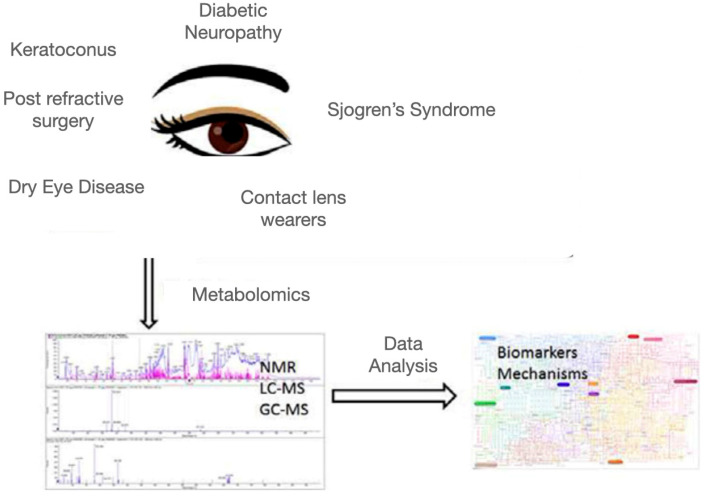
Metabolomics application to corneal diseases to identify new biomarkers or elucidate data for understanding metabolic pathways.

**Figure 2 metabolites-13-00380-f002:**
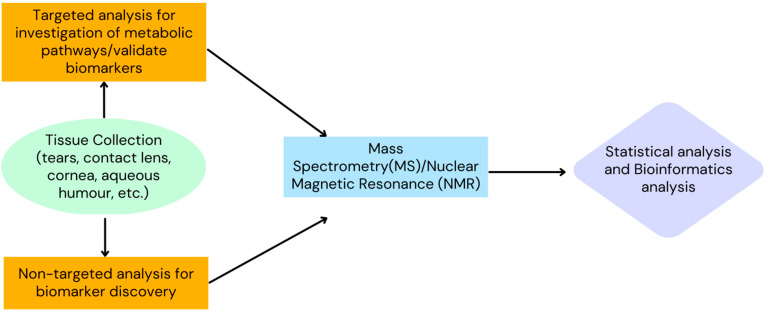
Flowchart depicting the methodology of corneal metabolomics.

**Table 1 metabolites-13-00380-t001:** Literature review of the metabolomic studies performed in corneal diseases.

Author	Metabolomic Assessment Method	Sample Source	Findings	Clinical Implications
Dry Eye Disease (DED)				
Lee et al. [51]	Nuclear magnetic resonance (NMR)	Rat tears, plasma, and urine	There was an increase in pro-inflammatory cytokines, such as IL-6, IL-1β, and TNF-α. 2-hydroxybutyrate, citrate and succinate, which play key roles in inflammatory pathways, had elevated levels in plasma.	Identified metabolites may serve as potential biomarkers for DED.
Chen et al. [10]	Matrix-assisted laser desorption ionization mass spectrometry imaging (MALDI-MSI)	Cornea, conjunctiva, and aqueous humor from rats	Glycerophospholipid and phenylalanine metabolism were altered, implicating their roles in signal transduction and tear film stability.Increase in betaine suggests early cytoprotective mechanism against dry eye.	MALDI-MSI can accurately analyze the different metabolic responses of complex eye components in DED, and can potentially be used to individualize treatment.
Quartieri et al. [43]	NMR	CBS and PBS eye drops	Metabolites measured in CBS and PBS eye drops were higher than tears. However, CBS drops contained higher amounts of myoinositol, choline, glutamine, creatine, and beta-hydroxybutyrate, which serve as anti-oxidants, and work in pathways to ameliorate DED.	Serum eye drops contain important metabolites to influence cellular signaling in hopes of alleviating DED.
Galbis-Estrada et al. [55]	NMR	Tears	There was an increase in choline after essential polyunsaturated fatty acids (EPUFA) supplementation—this stabilizes the tear film and has anti-inflammatory effects on ocular surface.	Changes in tear metabolic profile of DED can be modified by oral supplementation of antioxidants and EPUFAs.
Sjogren’s Syndrome				
Urbanski et al. [57]	Liquid chromatography—mass spectrometry (LC-MS)	Tears	9 metabolites could be used to distinguish PSS and DED.	Metabolomic signature of tears could distinguish PSS from DED.
Contact Lens Wear				
Li et al. [66]	LC-MS	Lenticules extracted via SMILE from soft contact lens-wearing patients	Upregulation of short chain organic acids indicate decreased respiration of glucose and switch to anaerobic respiration process in patients who wore contact lenses.	In the corneal stroma, there is significant change in energy metabolism in the corneal stroma after wearing soft contact lenses.
Refractive Surgery				
Li et al. [13]	LC-MS	Lenticules extracted from SMILE	There was a higher concentration of inflammatory-related metabolites and lower anti-oxidants in older patients. This influenced rates of corneal wound healing.	Corneal wound healing and corneal nerve recovery after SMILE was significantly affected by age.
Keratoconus				
Karamichos et al. [75]	LC-MS	Human corneal keratocytes (HCKs), fibroblasts (HCFs), and keratoconus cells (HKCs) cultured in 2D and 3D in vitro systems	Lactate levels and lactate/malate and lactate/pyruvate ratios were elevated in HKCs, while arginine and GSH/GSSG ratios were reduced, indicative of oxidative stress.	Future studies may help to identify novel pathways that may lead to metabolic therapies for keratoconus.
Wojakowska et al. [77]	Gas chromatography–MS	Corneal buttons	Downregulation of unsaturated fatty acids such as linoleic acid indicate reduced ability for cornea repair due to their anti-inflammatory effects. Succinic acid, playing an important role in tricarboxylic acid cycle, was downregulated.	Metabolomic signatures indicate oxidative stress and inflammatory reactions are involved in the development of keratoconus.
Kryczka et al. [78]	NMR and high-performance liquid chromatography	Corneal buttons from cadavers and keratoconic corneas	No major differences in metabolic contents between older, normal corneas and keratoconic corneas.	Young keratoconic corneas are biochemically like older normal corneas, which may indicate accelerated aging of the cornea in keratoconus.
Mckay et al. [83]	LC-MS	HKCs in a 3D in vitro construct	Lower cytoplasmic arginine and spermidine levels in KC constructs compared with healthy controls. Arginine supplementation led to a robust increase in cytoplasmic arginine, ornithine, and spermidine levels in controls only, and a significant increase in type I collagen secretion in HKC constructs.	Arginine supplementation may support increased collagen type I secretion by HKCs.
Saglik et al. [86]	LC-MS	Tears	N-acetyl-L-aspartic acid, which is known to have anti-oxidant effects, showed the greatest percentage increase after CXL.3-OH butyric acid showed the greatest decrease, suggesting improvement in glucose metabolism after CXL.	Metabolomic studies of tears could derive new understanding of processes in the follow-up period after CXL.
Snytnikova et al. [79]	NMR and LC-MS	Corneal buttons and aqueous humor from KC patients and normal corneas from cadavers	Levels of glucose in KC cornea was almost tenfold lower than in aqueous humor, demonstrating glucose consumption in vivo. Enhanced levels of citrate and acetate in KC corneas relate to oxidative stress, and is supported by low values of GSH/GSSG ratio in KC corneas.	Oxidative stress may be involved in keratoconus pathology.
Diabetic Corneas				
Brunmair et al. [89]	LC-MS	Tears	Upregulation of amino acids in tears is thought to act via the same pathways as insulin such as mammalian target of rapamycin (mTOR) and its downstream targets which act via the same pathways as insulin.Taurine accumulation in tears could indicate a stressed state.	High number of identified molecule markers may support disease development prediction, preventative approaches, personalized patients’ treatments, and monitoring treatment efficacy.
Priyadarsini et al. [97]	LC-MS	Cadaver cornea buttons	T1DM showed upregulation of kynurenic acid, suggesting dysregulation of kynurenine metabolism. Kynurenic acid is known to have anti-oxidant, anti-inflammatory, and anti-proliferative properties.	The kynurenine pathway is a potential therapeutic target to prevent T1DM-related complications to the eye.
Whelchel et al. [103]	LC-MS	In vitro 3D non-innervated constructs vs innervated constructs	Addition of neuron-supported pathways such as pyrimidine metabolism, glycerol phosphate shuttle, electron transport chain, and glycolysis.Myoinositol dysregulation was related to neuronal dysfunction in diabetic corneas.	Corneal innervation has impact on the metabolism of diabetic corneal stroma.

## Data Availability

Data is available from the individual cited articles. No new data was created.

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
