# Peer review of "Metabolomics in Corneal Diseases: A Narrative Review from Clinical Aspects"

_metabolites, 2023, doi:10.3390/metabo13030380_

Round 1

Reviewer 1 Report

This is well written article "Metabolomics in Corneal Diseases'' authors have tried to incorporate important information that may benefits the future researcher trying to work on eye diseases. I have few concerns....

1. Why this review article showing as a protocol?

2. Authors have tried to cover several metabolomics studies, but this manuscript could have been more informative with flow chart explaining the methodology of corneal metabolomics.

3. Figures are missing from current article, but they can increase the impact of manuscript. i.e., NMR spectra obtain from corneal sample (can use from previously published paper).

4.  Principle of metabolomic experiment (line 52) and Metabolism and metabolomic profiles in corneas (line 72) needs more references.

5.  line 115-117 needs references ....

Reviewer 2 Report

This manuscript has following shortcomings that need critical attention:

1.      Title should be modified. It is quite simple at its present form.

2.      In NMR technique, type of detector plays a key role for the detection of component of interest. But in case of metabolomics of eye diseases, which type of detector is preferably used?

3.      In case of hypertension, sodium intakes are generally prohibited. But in case of eye disease in hypertensive condition, what would be the fate of Na+/K+ active ion pump for the transport of water across the across the endothelium during corneal diseases?

4.      What and how many types of metabolites have been detected till date from the tears, cornea and aqueous humour of eyes suffered in various types of eye diseases?

5.      Techniques used in the metabolomics profiles of eye diseases have not been described.

6.      T2DM plays a crucial role in eye diseases and in association with CVDs, T2DM becomes more worse particularly for eyes. This aspect is  also missing in this manuscript.

7.      The scientific language of this manuscript is very poor. Extensive revision is required to improve it.

Reviewer 3 Report

This article is a really interesting review concerning the metabolomic studies undertaken in the cornea and in several diseases.

This review is really well-written, well-structured and clearly describes this challenging topic.

In my opinion, it could be accepted in the present form.

Round 2

Reviewer 2 Report

Authors have addressed all the comments. I have no further comments